# Impacts of marine instability across the East Antarctic Ice Sheet on Southern Ocean dynamics

Steven J. Phipps[1,2], Christopher J. Fogwill[1,3], and Christian S. M. Turney[1]

[1]Climate Change Research Centre, School of Biological, Earth and Environmental Sciences, UNSW Australia, Sydney, NSW 2052, Australia.
[2]Institute for Marine and Antarctic Studies, University of Tasmania, Hobart, TAS 7001, Australia.
[3]PANGEA Research Centre, UNSW Australia, Sydney, NSW 2052, Australia.

*Correspondence to:* Steven J. Phipps (Steven.Phipps@utas.edu.au)

**Abstract.** Recent observations and modelling studies have demonstrated the potential for rapid and substantial retreat of large sectors of the East Antarctic Ice Sheet (EAIS). This has major implications for ocean circulation and global sea level. Here we examine the effects of increasing meltwater from the Wilkes Basin, one of the major marine-based sectors of the EAIS, on Southern Ocean dynamics. Climate model simulations reveal that the meltwater flux rapidly stratifies surface waters, leading to a dramatic decrease in the rate of Antarctic Bottom Water formation. The surface ocean cools but, critically, the Southern Ocean warms by more than $1°C$ at depth. This warming is accompanied by a Southern Ocean-wide 'domino effect', whereby the warming signal propagates westward with depth. Our results suggest that melting of one sector of the EAIS could result in accelerated warming across other sectors, including the Weddell Sea sector of the West Antarctic Ice Sheet. Thus localised melting of the EAIS could potentially destabilise the wider Antarctic Ice Sheet.

## 1 Introduction

The Fifth Assessment Report of the Intergovernmental Panel on Climate Change highlights the fact that current and future anthropogenic greenhouse gas emissions are likely to affect the Earth's climate for millennia to come (Collins et al., 2013). One major uncertainty relates to how the marine-based sectors of the Antarctic Ice Sheet will respond to future climate change, and particularly to changes in Southern Ocean circulation (Lenton et al., 2008; Pritchard et al., 2012; Vaughan et al., 2013). Given that large sectors of West and East Antarctica have been demonstrated to be sensitive to Southern Ocean circulation, there is an urgent need to understand potential ice sheet-ocean feedbacks.

Of particular concern are the substantial sectors of the East Antarctic Ice Sheet (EAIS) that are underlain by extensive marine-based subglacial basins. These have an associated global sea level rise potential of ∼14m and have been shown to be vulnerable to marine ice sheet instability (e.g. Mengel and Levermann, 2014). The potential instability of marine-based sectors of the EAIS adds considerable uncertainty to projections of future global sea level rise, suggesting that current upper estimates of an increase of ∼1m over the coming century may be conservative (Joughin et al., 2014; Golledge et al., 2015).

Recent studies have highlighted the sensitivity of the EAIS to changes in ocean dynamics during climatically warm periods, including those that have occurred in the past and those that are projected to occur in future. Examining the climate of the

Last Interglacial (∼135,000–116,000 years ago), Fogwill et al. (2014b) conclude that the EAIS may have made a substantial contribution towards interglacial sea levels. Climate modelling suggests that a southward migration of the Southern Hemisphere westerly winds drove warm Circumpolar Deep Water onto the continental shelves, causing pervasive ocean warming at depth along the margins of the EAIS. This would have enhanced basal melting, potentially triggering retreat of the EAIS through
increased mass flux. A similar and ongoing migration of the westerly winds, driven by anthropogenic forcings, has taken place over recent decades and may continue into the future (e.g. Arblaster and Meehl, 2006; Arblaster et al., 2011). Examining both past and future warm periods, Golledge et al. (2015) and DeConto and Pollard (2016) also demonstrate the highly dynamic nature of the EAIS, and particularly its sensitivity to warmer-than-present ocean temperatures. Using continental-scale ice sheet models, they conclude that future anthropogenic warming has the potential to cause partial or complete melting of the
floating ice shelves around Antarctica. Such a scenario would result in a retreat of the major subglacial basins and an ongoing commitment towards global sea level rise.

While these previous studies have highlighted the potential sensitivity of the EAIS to warmer-than-present conditions, and therefore to future anthropogenic climate change, they have only considered one aspect of ice–ocean coupling: the impact of changes in the oceanic circulation on the ice sheet. Fogwill et al. (2015) consider instead the impact of changes in the ice
sheet on the ocean, and show that the dynamics of the Southern Ocean are sensitive to increases in meltwater flux from the West Antarctic Ice Sheet (WAIS). This raises the possibility of regional ice sheet–ocean feedback loops that could accelerate any anthropogenically-triggered melting of sectors of the WAIS. Hattermann and Levermann (2010) also identify three distinct feedback loops whereby basal melting of the Antarctic Ice Sheet influences the circulation of the Southern Ocean. The dominant feedback loop involves local cooling; this loop is negative. A secondary feedback loop involves acceleration of the subpolar
gyres; this loop is positive. A minor additional feedback loop involves weakening of the Antarctic Circumpolar Current (ACC); this loop is negative.

In this study, we explore such feedback loops from the perspective of the EAIS. We present a suite of model simulations that consider idealised scenarios corresponding to the collapse of the Wilkes Basin, and analyse the potential impact of the meltwater flux on the dynamics of the Southern Ocean over the coming centuries and millennia. The Wilkes Basin is selected
for the current study as it is known to be particularly vulnerable to marine ice sheet instability, with the potential to raise global sea levels by ∼3–4m (Mengel and Levermann, 2014). It also lies alongside the Mawson Gyre (Fogwill et al., 2014b), which is connected to the other sub-polar gyres in the Southern Ocean (McCartney and Donohue, 2007) and therefore provides a mechanism by which meltwater may be transmitted around the continent.

To better understand both present and potential future interactions between the EAIS and the ocean around Antarctica, we
undertake a number of simulations using a fully-coupled climate system model. We use these simulations to investigate the response of the Southern Ocean to meltwater input from specific locations along the George V Coast, which fringes the Wilkes Basin in East Antarctica (Fig. 1). Previous climate modelling studies have highlighted the sensitivity of the simulated regional and global impacts to the precise location of the freshwater input (Weaver et al., 2003; Fogwill et al., 2015). By applying the freshwater fluxes at different locations within our model, we are able to assess the sensitivity of our results to the details of the
experimental design.

The model simulations are described in Sect. 2. The response of the Southern Ocean under a default scenario representing the collapse of the Wilkes Basin is assessed in Sect. 3. The sensitivity of our results to the precise location of the meltwater input is explored in Sect. 4. The response to meltwater input is compared and contrasted with the response to anthropogenic forcing in Sect. 5. Finally, the results are discussed and conclusions presented in Sect. 6.

## 2  Methods

We use the CSIRO Mk3L climate system model version 1.2 in this study. The model comprises fully interacting atmosphere, ocean, land surface and sea ice components, and is designed for long-term climate simulations (Phipps et al., 2011, 2012). The ocean model has a horizontal resolution of $2.8° \times 1.6°$ and 21 vertical levels, while the atmosphere model has a horizontal resolution of $5.6° \times 3.2°$ and 18 vertical levels. We use CSIRO Mk3L here because it includes full dynamic complexity in its sub-models, and yet is sufficiently fast to allow us to explore the long-term evolution of the Southern Ocean to meltwater fluxes. Although the computational efficiency is achieved at the cost of a relatively coarse spatial resolution, the model has demonstrated utility for simulating past and future changes in the climate system. This includes the millennial-scale response to natural and anthropogenic forcings (Phipps et al., 2012, 2013) and the long-term response of the Southern Ocean to enhanced Antarctic meltwater input (Fogwill et al., 2015).

We employ an ensemble modelling approach. For each experiment, the model is integrated three times. Each of these simulations is identical, except for the fact that it is initialised from a different year of a pre-industrial control simulation. Such an approach allows us to better distinguish between natural internal variability and the forced response of the climate system to the meltwater fluxes. The results reported here represent the ensemble mean for each experiment, with any anomalies being calculated relative to the pre-industrial control simulation. The statistical significance of the anomalies is determined using a Student $t$ test, under the null hypothesis that the climate of the hosing simulations is identical to the climate of the control simulation.

We focus on three idealised scenarios, each of which represents a hypothetical collapse of the Wilkes Basin: WILKES, WEST and EAST. The locations of meltwater input within these experiments are shown in Fig. 2. In all of the experiments, a freshwater flux of 0.048Sv is applied for 900 years. This is based on the estimate of Mengel and Levermann (2014), and is equivalent to an increase of ~3.8m in global sea level. While the precise mechanisms that may trigger instability across the extensive Wilkes Basin remain a subject of debate (Mengel and Levermann, 2014; Golledge et al., 2015; Pollard et al., 2015), such a rapid and substantial addition to global sea level in future is hypothetically possible. The meltwater that is added to the ocean is assumed to have the same temperature as the ambient seawater, and thus there is no heat flux associated with the meltwater input. Once the freshwater flux has been applied, the experiments are then integrated for a further 600 years to allow the climate system to recover.

In its current configuration, the Wilkes Basin is drained through the Ninnis and Cook ice streams along the George V Coast (Mengel and Levermann, 2014). The meltwater input within the model simulations is therefore implemented as a flux into the Southern Ocean in this region (Figures 1 and 2). However, the Wilkes ice sheet rests on two deep troughs that lie below sea

level (Mengel and Levermann, 2014; Pollard et al., 2015). In the event of a substantive collapse and retreat of the Wilkes Basin, discharge into the Southern Ocean would therefore occur at a location up to $\sim$800km south of the current position (Mengel and Levermann, 2014). We do not take this into account in our experiments, maintaining a constant location for the freshwater input throughout. Neither do we explicitly account for calving of icebergs, assuming instead that meltwater is delivered immediately

and locally to the Southern Ocean.

    Experiments WEST and EAST are identical to WILKES, except that the hosing region is displaced to the west and east, respectively (Fig. 2). A critical difference between these two additional experiments is that, in WEST, the hosing region is situated in the open Southern Ocean while, in EAST, it is located in an embayment on the model grid. These experiments assess the sensitivity of our results to the precise location of the freshwater input, and therefore assess the sensitivity of our

results to our assumptions regarding the location of meltwater delivery from the Wilkes Basin.

    In this study, we wish to isolate the effects of enhanced meltwater input from the Wilkes Basin and to explore potential feedback loops. We therefore intentionally neglect the meltwater that is likely to come from other sectors of the Antarctic Ice Sheet under future climate change scenarios. We also use a constant atmospheric $CO_2$ concentration of 280ppm in all of our hosing experiments. However, elevated atmospheric $CO_2$ concentrations can have substantial implications for both sea ice

production and AABW formation (e.g. Swingedouw et al., 2008; Phipps et al., 2012; Collins et al., 2013; Joughin et al., 2014). We therefore run an additional experiment, 4CO2, in which the atmospheric $CO_2$ concentration is quadrupled. Starting from the pre-industrial level of 280ppm, the $CO_2$ concentration is increased at 1% per year until it reaches 1120ppm after 140 years. It is then held constant thereafter. There is no freshwater input into the Southern Ocean in this experiment.

    Past climate modelling studies have employed intermediate complexity models (e.g. Weaver et al., 2003) or have applied

freshwater fluxes across large sectors of the Southern Ocean (e.g. Swingedouw et al., 2009; Menviel et al., 2010). However, other recent studies have shown that Antarctic meltwater input can be highly localised (Shepherd et al., 2004; Rye et al., 2014), and yet still have important implications for the Southern Ocean circulation, atmospheric processes, sea ice regimes and potentially ice sheet dynamics (Pritchard et al., 2012; Bintanja et al., 2013; Rhein et al., 2013; Vaughan et al., 2013; Fogwill et al., 2015). Thus, while our experiments represent idealised scenarios, they nonetheless allow us to explore the response of

the Southern Ocean to localised meltwater input. This is critical for better understanding the evolution and response of the Southern Ocean to future meltwater inputs that may be highly spatially localised.

## 3   Marine instability across the Wilkes Basin

The rate of Antarctic Bottom Water (AABW) formation in experiment WILKES is shown in Fig. 3a. The mean rate of AABW formation in the control simulation is 6.8Sv, which is close to the observational estimate of 8.1–9.4Sv (Orsi et al., 1999).

Upon the commencement of the freshwater hosing, there is a rapid reduction of $\sim$20% in the rate of AABW formation. It then remains in a weakened state throughout the hosing phase. There is a rapid recovery as soon as the freshwater hosing ceases, although it takes several centuries before the rate of AABW formation returns to the same level as in the control simulation. Considerable low-frequency variability is apparent within each simulation, even after the application of a 100-year running

mean filter. This highlights the merits of an ensemble modelling approach, which allows us to better isolate the simulated response of the climate system to the meltwater input.

For reference, the pre-industrial climatology of the model is presented in Fig. 4. Consistent with observations (e.g. Orsi et al., 1999), simulated deep water formation occurs in the Weddell Sea, the western Ross Sea and, to a lesser extent, in the region around the Amery Basin (Fig. 4f). However, deep water is created through deep convection in the open ocean, rather than forming over the continental shelves. Other climate system models, including the higher-resolution models that participated in the Coupled Model Intercomparison Project Phase 5 (CMIP5), display a similar bias (Heuzé et al., 2013; Meijers, 2014). This reflects the difficulty of capturing the fine-scale interactions that drive the formation of AABW (Heuzé et al., 2013; Meijers, 2014). The model simulates a Weddell Gyre with a strength of 30Sv, but does not resolve a gyre in the Ross Sea (Fig. 4h).

The response of the Southern Ocean to the freshwater input in experiment WILKES is presented in Fig. 5. The changes are shown as the ensemble-mean anomalies during the final 100 years of the hosing phase, relative to the equivalent years of the pre-industrial control simulation. Sea surface temperatures (SSTs) increase by ∼1°C off the coast of Wilkes Land (Fig. 5a). Weaker warming is also apparent in other coastal regions, particularly in the Bellinghausen, Amundsen and Ross Seas. Cooling occurs over the open Southern Ocean, and is strongest to the north of the Ross Sea. A similar temperature signal is apparent at a depth of 200–400m, with the greatest increase of 1.2°C occurring at 122°E along the coast of Wilkes Land (Fig. 5b). However, the warming at this depth is more evenly distributed around Antarctica. Intriguingly, the warming signal is also found deeper within the water column, propagating westwards around the coast of Antarctica with depth (Fig. 5c–d). At a depth of 400-700m, the greatest temperature increase of 0.9°C occurs at 75°E, at the mouth of the Amery Ice Shelf. The strongest warming at a depth of 700-1000m is also found in this region, but a temperature increase of a similar magnitude is observed along the coast of Dronning Maud Land as well.

Thus the response of the model is characterised by two key features: firstly, there is a near-instantaneous reduction in AABW formation as soon as the freshwater hosing is applied; secondly, there is a warming signal along the coast of the EAIS that propagates westwards with depth.

To explore the dynamical mechanisms that give rise to this behaviour, the change in the sea surface salinity (SSS) is shown in Fig. 5e. The meltwater input causes a strong reduction in the SSS throughout the hosing region. However, the coastal currents (Fig. 6) also carry the fresher surface waters westward, with negative SSS anomalies propagating as far west as the Weddell Sea. This freshening reduces the density of the surface waters, increasing the vertical stratification of the water column and reducing convective depth (Fig. 5f; the convective depth is defined here as the maximum depth over which the water column is well mixed through convection of dense surface waters). The largest changes occur in the regions of deep water formation within the model (Fig. 4f), particularly in the Weddell Sea and, to a lesser extent, in the western Ross Sea.

Remarkably, we therefore find that the largest changes in convective depth occur on the opposite side of the continent from the region of freshwater input. Furthermore, this outcome occurs despite the fact that the salinity signal in the Weddell Sea is weak; the average decrease in SSS between 37°W and 8°W, where the greatest reduction in convective depth occurs, is only 0.13psu. This demonstrates that the Weddell Sea is extremely sensitive to freshwater input within the model and can

be significantly impacted by melting on the other side of the continent, as a result of the surface freshening being carried westwards by the coastal currents.

The precise mechanism whereby a surface freshening can give rise to warming at depth is explored by Fogwill et al. (2015). An enhanced meltwater flux into the Southern Ocean stratifies the surface waters, reducing vertical mixing and hence reducing the exchange of cold surface waters with the underlying warmer waters. This leads to cooling at the surface and warming at depth. We see the same mechanism operating in our simulations: as the fresher surface waters propagate westwards, we find a reduction in convection, accompanied by a reduction in temperature at the surface and an increase in temperature at depth. However, an exception to this pattern is found along the coast of Wilkes Land, where sea surface temperatures are warmer. This is a region where there is no vertical mixing within the model (Fig. 4f). Instead, the warming appears to be due to local sea ice feedbacks, with a marked reduction in sea ice cover in this region (Fig. 5g).

The strength of the ACC in experiment WILKES is shown in Fig. 3b. The mean strength of the ACC in the control simulation is 175Sv, which is stronger than the observational estimate of 136.7±7.8Sv (Cunningham et al., 2003). There is a small reduction of ∼2Sv upon the commencement of freshwater hosing, although this reduction does not persist until the end of the hosing phase. A weak reduction in the strength of the ACC in response to ice sheet melting is consistent with the results of Hattermann and Levermann (2010). The changes in the barotropic streamfunction (Fig. 5h) reveal a reduction in the strength of the Weddell Gyre, but a positive gyre-like anomaly in the Ross Sea. This contrasts with Hattermann and Levermann (2010), who find a strengthening of both the Weddell and Ross Sea Gyres in response to ice sheet melting.

## 4   Sensitivity to hosing region

The rates of AABW formation for experiments WEST and EAST are shown in Fig. 3a. The behaviour of both experiments is broadly similar to WILKES. There is a rapid and sustained reduction upon the application of freshwater hosing, followed by a rapid recovery once the hosing ceases. However, there is slightly stronger suppression of AABW formation in WEST than in WILKES or EAST.

Consistent with the greater decrease in AABW formation, the surface warming is strongest in WEST, with an increase of up to 1.9°C in SST (Fig. 7a). This is noticeably stronger than the maximum warming of 1.2°C in WILKES. In contrast, the surface temperature response is noticeably weaker in EAST than in WILKES, with maximum warming of only 0.9°C (Fig. 8a). This pattern is mirrored in the temperature response at depth (Figures 7b–d and 8b–d). Both WEST and EAST simulate a similar pattern of warming to WILKES, including the westward propagation of the temperature changes. However, the magnitude of the changes is largest in WEST and smallest in EAST.

Examination of the surface freshening reveals the explanation for this difference in behaviour between the experiments. In WEST, where the freshwater is applied to the open ocean, the surface freshening propagates further around the continent than in WILKES (Fig. 7e). In contrast, in EAST, where the freshwater is applied to an embayment on the model grid, the surface freshening remains much more localised (Fig. 8e). Over the eastern part of the Weddell Sea, between 37°W and 8°W,

the average decrease in SSS is 0.22psu in WEST but 0.13psu in EAST. The resulting suppression of convection is therefore stronger (Figures 7f and 8f).

## 5   Comparison with the response to anthropogenic forcing

Experiment 4CO2 allows us to compare and contrast the response to meltwater input with the response to anthropogenic forcing. Fig. 3a reveals a strong and persistent collapse in the rate of AABW formation in response to a quadrupling of the atmospheric $CO_2$ concentration within experiment 4CO2. This can be attributed to the large reduction in sea ice cover (Fig. 9g; see also Phipps et al., 2012). The freshwater hosing experiments WILKES, WEST and EAST also feature a reduction in the rate of AABW formation, indicating that melting of the EAIS might be expected to amplify the anthropogenic signal.

There is a strong and persistent increase in the strength of the ACC (Fig. 3b). This is in contrast to the small reduction in strength in the freshwater hosing experiments, indicating that ice sheet melting might act as a weak negative feedback in this case. This result is consistent with the findings of Hattermann and Levermann (2010). The changes in the barotropic streamfunction indicate strengthening of the Weddell Gyre and a strong positive gyre-like anomaly in the Ross Sea. The change in the Weddell Gyre contrasts with the weakening in the freshwater hosing experiments, while the changes in the Ross Sea are consistent. Thus ice sheet melting might be expected to act as a negative feedback in the Weddell Gyre, but a positive feedback in the Ross Sea. This constrasts with the results of Hattermann and Levermann (2010), where ice sheet melting acts as a positive feedback in both sectors.

The response of the Southern Ocean is presented in Fig. 9. The changes shown are the ensemble-mean anomalies during the first 100 years after the atmospheric $CO_2$ concentration has stabilised (i.e. 141–240 years after the start of the experiment), relative to the equivalent years of the pre-industrial control simulation. Although SST increases by up to 5.4°C along the coast of Antarctica (Fig. 9a), the maximum warming adjacent to the coast decreases with depth to 2.8°C at 200–400m (Fig. 9b), 2.2°C at 400–700m (Fig. 9c), and 2.3°C at 700–1000m (Fig. 9d).

These figures compare with warming of up to ∼1°C in response to a collapse of the Wilkes Basin within experiment WILKES. In specific locations, the responses to the two forcings can be similar in magnitude. For example, at the mouth of the Amery Ice Shelf at 72°E, the temperature increase at a depth of 700–1000m is 1.0°C in 4CO2 and 0.8°C in WILKES. Ice sheet melting therefore acts as a strong positive feedback, amplifying the warming adjacent to the grounding lines of the Antarctic Ice Sheet.

Overall, this comparison reveals that melting of a single sector of the EAIS has the potential to act as a positive feedback that can amplify the response to anthropogenic forcing. Our experiments reveal that ice sheet melting might be expected to act as a positive feedback on the changes in AABW formation, the Ross Sea Gyre and, in particular, on warming of the ocean at depth. Only in the case of the ACC and the Weddell Gyre might ice sheet melting be expected to act as a weak negative feedback.

## 6 Discussion and conclusions

Using a coupled climate system model, we have found that deep water formation in the Weddell Sea is highly sensitive to freshwater input from a key sector of the EAIS. A previous modelling study has also found that overturning in the Weddell Sea is sensitive to changes in the salinity of the surface ocean (Galbraith et al., 2011). However, we establish here that remote

freshwater input is sufficient to trigger a reduction in Weddell Sea convection. Melting of the EAIS can therefore lead to a reduction in AABW formation, as a result of fresher surface waters being carried westwards by the coastal currents. This result has significant implications for the response of the Southern Ocean to the melting of sectors of the Antarctic Ice Sheet.

The warming at increasing depth around the coast can be attributed to the coastal currents, giving rise to a 'domino effect'. According to this effect, the temperature changes propagate westwards around the coast of the Antarctic continent with in-

creasing depth, representing a positive feedback mechanism that has the potential to amplify melting around the continent. Indeed, we find that this mechanism occurs even without the freshwater melt from other sectors of the Antarctic Ice Sheet. The rapid recovery in AABW formation that we find within our experiments might therefore be considered conservative.

We speculate that an initial melting, in this instance off the coast of Wilkes Land, might set off a warming signal with increasing depth around the coast. This would have the potential to enhance melting along vulnerable grounding lines across

the EAIS, which would then result in additional warming and melting further to the west. Subsequent warming and melting might be expected even if the melting of the Wilkes subglacial basin later reduces in magnitude or ceases entirely. Thus destabilisation of large sectors of the EAIS could arise from warming and melting in just one area.

Furthermore, recent work has highlighted the sensitivity of the Weddell Sea sector of the WAIS to changes in local ocean circulation (Hellmer et al., 2012). This is exacerbated by the presence of steep reverse slope beds in regional ice streams (Ross

et al., 2012), making this sector particularly vulnerable to warming (Humbert, 2012; Fogwill et al., 2014a; Hillenbrand et al., 2014). We have established in this study that melting of the EAIS can lead to reduced convection and warming at depth in the Weddell Sea. The Weddell Sea sector of the WAIS is grounded at a depth of $\sim$1000-1200m below sea level (Hillenbrand et al., 2014). It is therefore vulnerable to the deep warming that we find in our experiments, suggesting that localised melting of one sector of the EAIS might be sufficient to destabilise at least one key sector of the WAIS as well.

Through a comparison of our results with an experiment in which the atmospheric $CO_2$ concentration is quadrupled, we have shown that anthropogenically-induced melting of the EAIS has the potential to act as a positive feedback on the $CO_2$-forced changes in AABW formation, the Ross Sea Gyre and, in particular, on the warming of the Southern Ocean adjacent to the grounding lines of the Antarctic Ice Sheet. Only in the case of the ACC and the Weddell Gyre might melting of the EAIS be expected to act as a weak negative feedback. The response of the Southern Ocean is sufficiently rapid that, on sub-centennial

timescales, it has the potential to exceed the magnitude of the initial response to enhanced atmospheric $CO_2$.

We have conducted a suite of experiments which reveal that, to first order, our conclusions are robust with regard to the location of freshwater input within the model. However, we also find that the magnitude of the simulated changes is sensitive to the precise location of the hosing region. This demonstrates that care must be taken when designing experiments which simulate the effects of ice sheet melt.

This study is based on a single climate system model, and therefore we cannot rule out the possibility that the results are simply a model artefact. In particular, one potential bias arises from the fact that CSIRO Mk3L forms AABW through open-ocean deep convection rather than over the continental shelves, consistent with other models (Heuzé et al., 2013; Meijers, 2014). Given the potential importance of the feedback loop that we have identified for future changes in the Antarctic Ice Sheet

and global sea level, similar experiments should be conducted using additional models. Climate system models with a high spatial resolution might be better able to resolve the fine-scale interactions that control the formation of AABW (Heuzé et al., 2013; Meijers, 2014). Nonetheless, we have highlighted the importance of incorporating ice sheet–ocean processes into the earth system models that are used to generate future climate projections.

*Author contributions.*  All the authors contributed towards the design of the study. SJP conducted the model simulations and analysed the

output. SJP led the writing of the manuscript, with contributions from all the authors.

*Acknowledgements.*  This research was supported under the Australian Research Council's *Laureate Fellowships* funding scheme (project ID FL10010019), *Future Fellowships* funding scheme (project ID FT120100004) and Special Research Initiative for the Antarctic Gateway Partnership (project ID SR140300001). It was undertaken with the assistance of resources provided at the Australian National University through the National Computational Merit Allocation Scheme supported by the Australian Government. It also includes computations using the Linux

computational cluster Katana supported by the Faculty of Science, UNSW Australia. The authors wish to acknowledge use of the Ferret program for analysis and graphics; Ferret is a product of NOAA's Pacific Marine Environmental Laboratory (http://ferret.pmel.noaa.gov/Ferret/). This research forms a contribution to the Australasian Antarctic Expedition 2013–2014 (www.spiritofmawson.com).

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

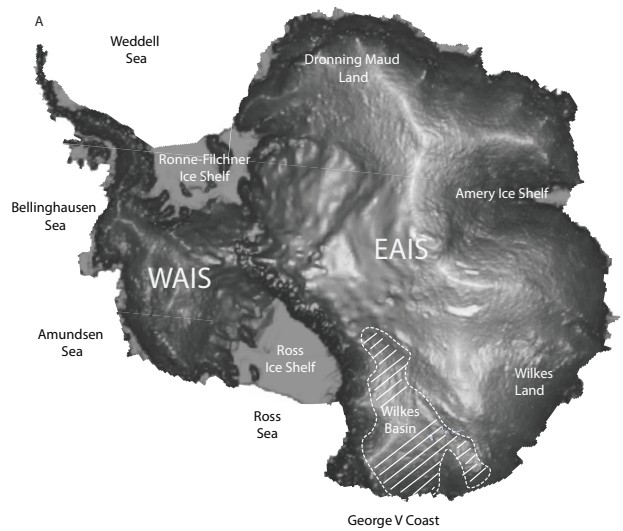

**Figure 1.** Map of the Antarctic Ice Sheet, highlighting the location of the Wilkes Basin.

Rye, C. D., Garabato, A. C. N., Holland, P. R., Meredith, M. P., Nurser, A. J. G., Hughes, C. W., Coward, A. C., and Webb, D. J.: Rapid sea-level rise along the Antarctic margins in response to increased glacial discharge, Nat. Geosci., 7, 732–735, doi:10.1038/ngeo2230, 2014.

Shepherd, A., Wingham, D., and Rignot, E.: Warm ocean is eroding West Antarctic Ice Sheet, Geophys. Res. Lett., 31, L23 402, doi:10.1029/2004GL021106, 2004.

Swingedouw, D., Fichefet, T., Huybrechts, P., Goosse, H., Driesschaert, E., and Loutre, M.-F.: Antarctic ice-sheet melting provides negative feedbacks on future climate warming, Geophys. Res. Lett., 35, L17 705, doi:10.1029/2008GL034410, 2008.

Swingedouw, D., Fichefet, T., Goosse, H., and Loutre, M.-F.: Impact of transient freshwater releases in the Southern Ocean on the AMOC and climate, Clim. Dynam., 33, 365–381, doi:10.1007/s00382-008-0496-1, 2009.

Vaughan, D. G., Comiso, J. C., Allison, I., Carrasco, J., Kaser, G., Kwok, R., Mote, P., Murray, T., Paul, F., Ren, J., Rignot, E., Solomina, O., Steffen, K., and Zhang, T.: Observations: Cryosphere, in: Climate Change 2013: The Physical Science Basis. Contribution of Working Group I to the Fifth Assessment Report of the Intergovernmental Panel on Climate Change, edited by Stocker, T. F., Qin, D., Plattner, G.-K., Tignor, M., Allen, S. K., Boschung, J., Nauels, A., Xia, Y., Bex, V., and Midgley, P., chap. 4, pp. 317–382, Cambridge University Press, 2013.

Weaver, A. J., Saenko, O. A., Clark, P. U., and Mitrovica, J. X.: Meltwater Pulse 1A from Antarctica as a trigger of the Bølling-Allerød warm interval, Science, 299, 1709–1713, doi:10.1126/science.1081002, 2003.

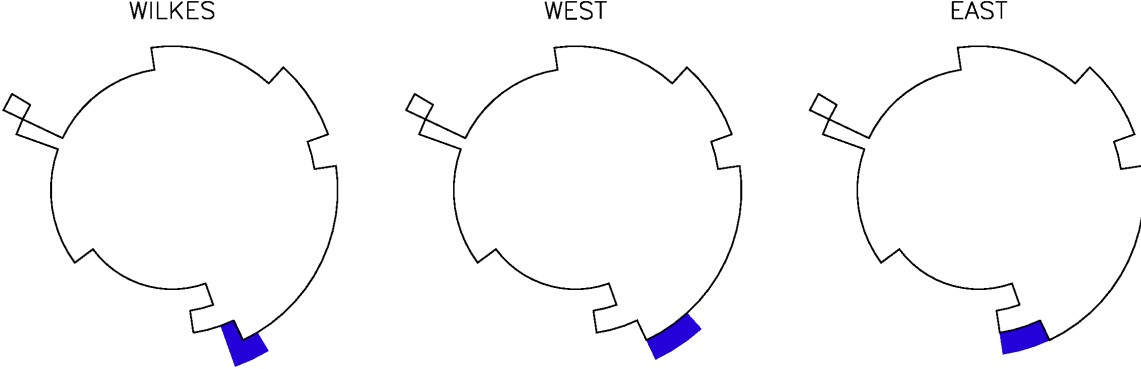

**Figure 2.** Locations of freshwater input in experiments WILKES, WEST and EAST.

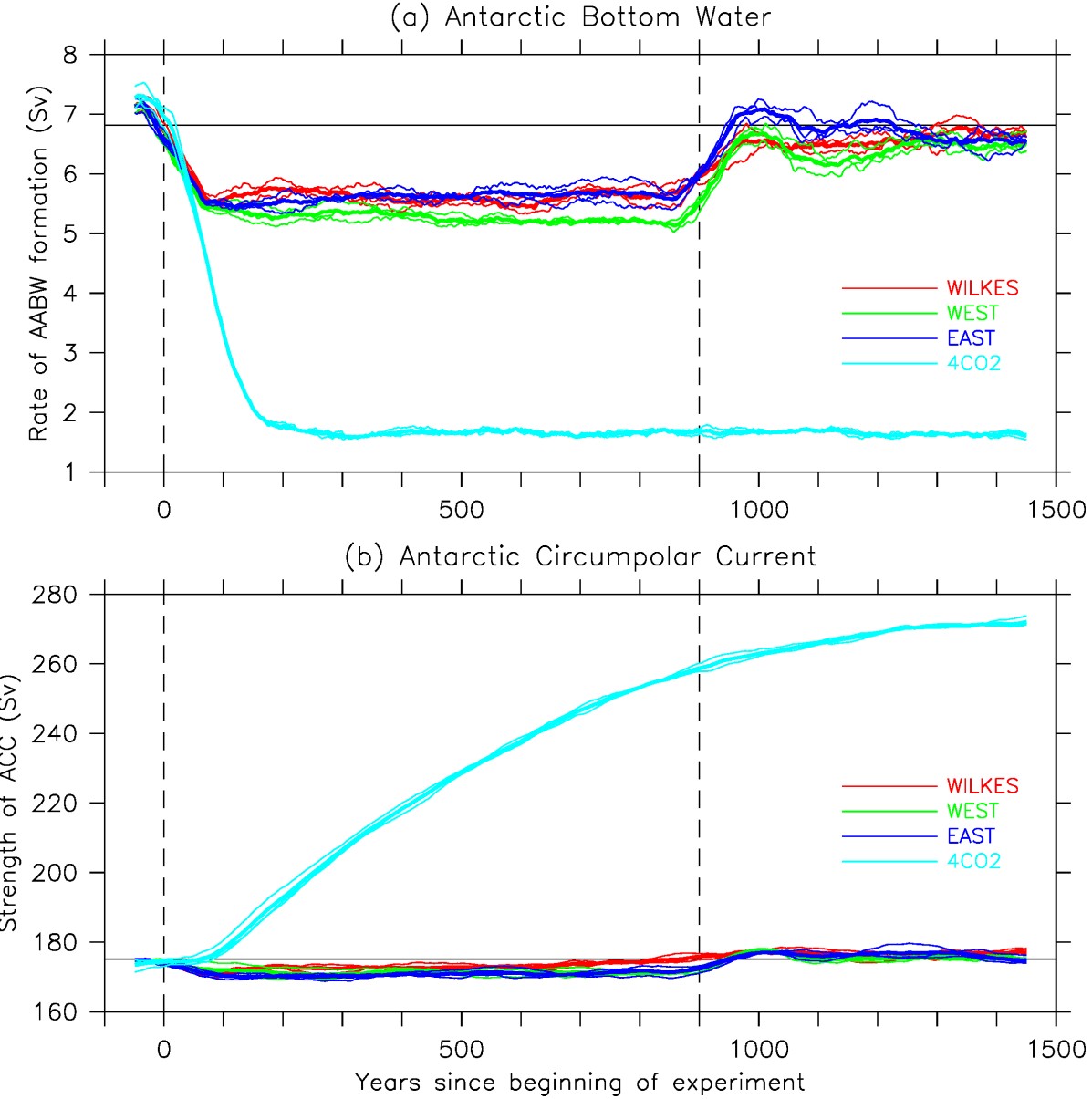

**Figure 3. (a)** The rate of AABW formation (Sv), and **(b)** the strength of the ACC (Sv), in experiments WILKES (red), WEST (green), EAST (dark blue) and 4CO2 (light blue). Thin lines indicate individual ensemble members; thick lines indicate the ensemble means. The values shown are 100-year running averages. Solid horizontal lines indicate the mean values for the control simulation. Dashed vertical lines indicate the years in which the freshwater hosing begins and ends.

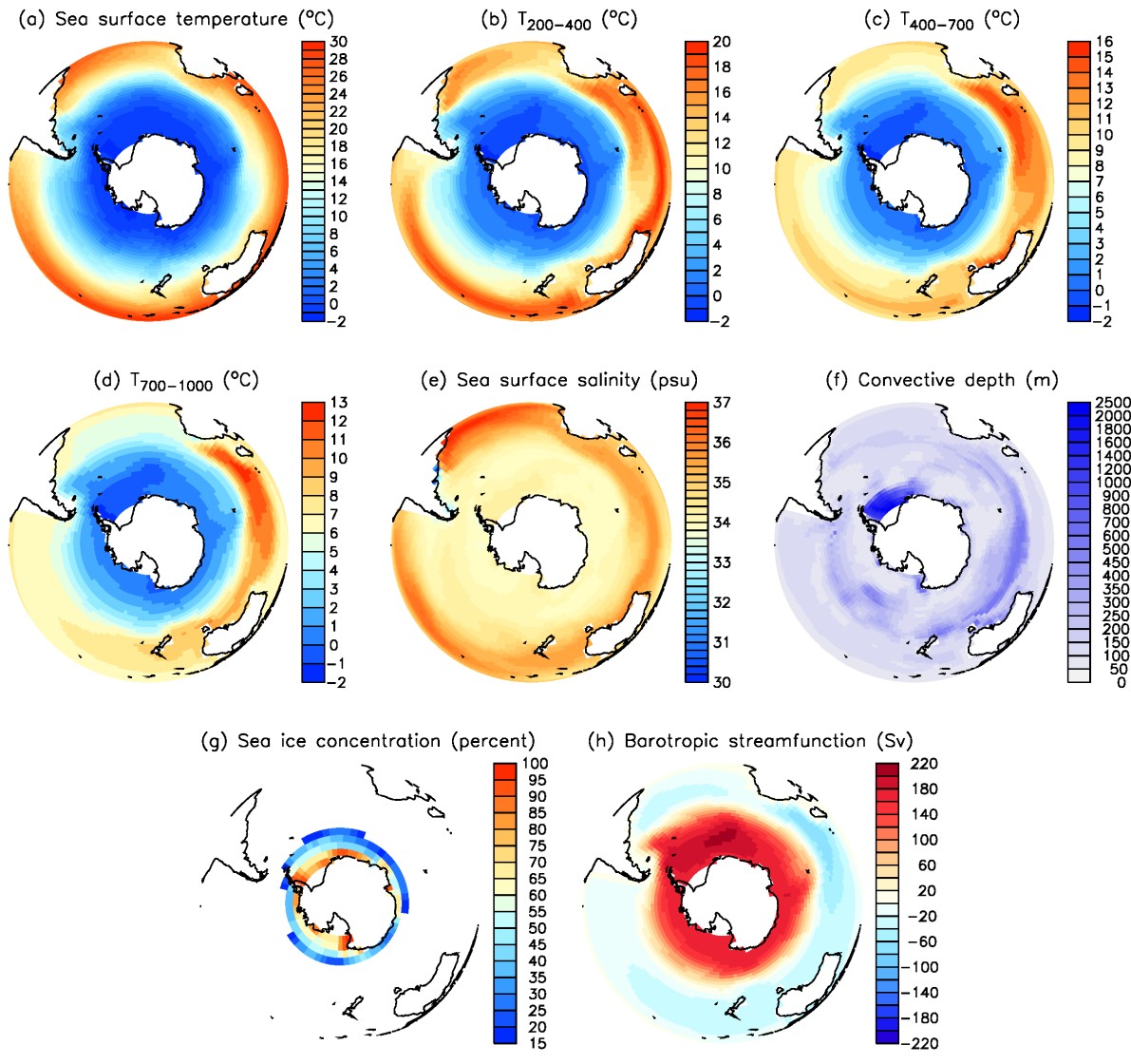

**Figure 4.** Annual-mean climatology for the pre-industrial control simulation: **(a)** sea surface temperature ($^\circ$C), **(b)** mean temperature at a depth of 200–400m ($^\circ$C), **(c)** mean temperature at a depth of 400–700m ($^\circ$C), **(d)** mean temperature at a depth of 700–1000m ($^\circ$C), **(e)** sea surface salinity (psu), **(f)** convective depth (m), **(g)** sea ice concentration (%), and **(h)** barotropic streamfunction (Sv).

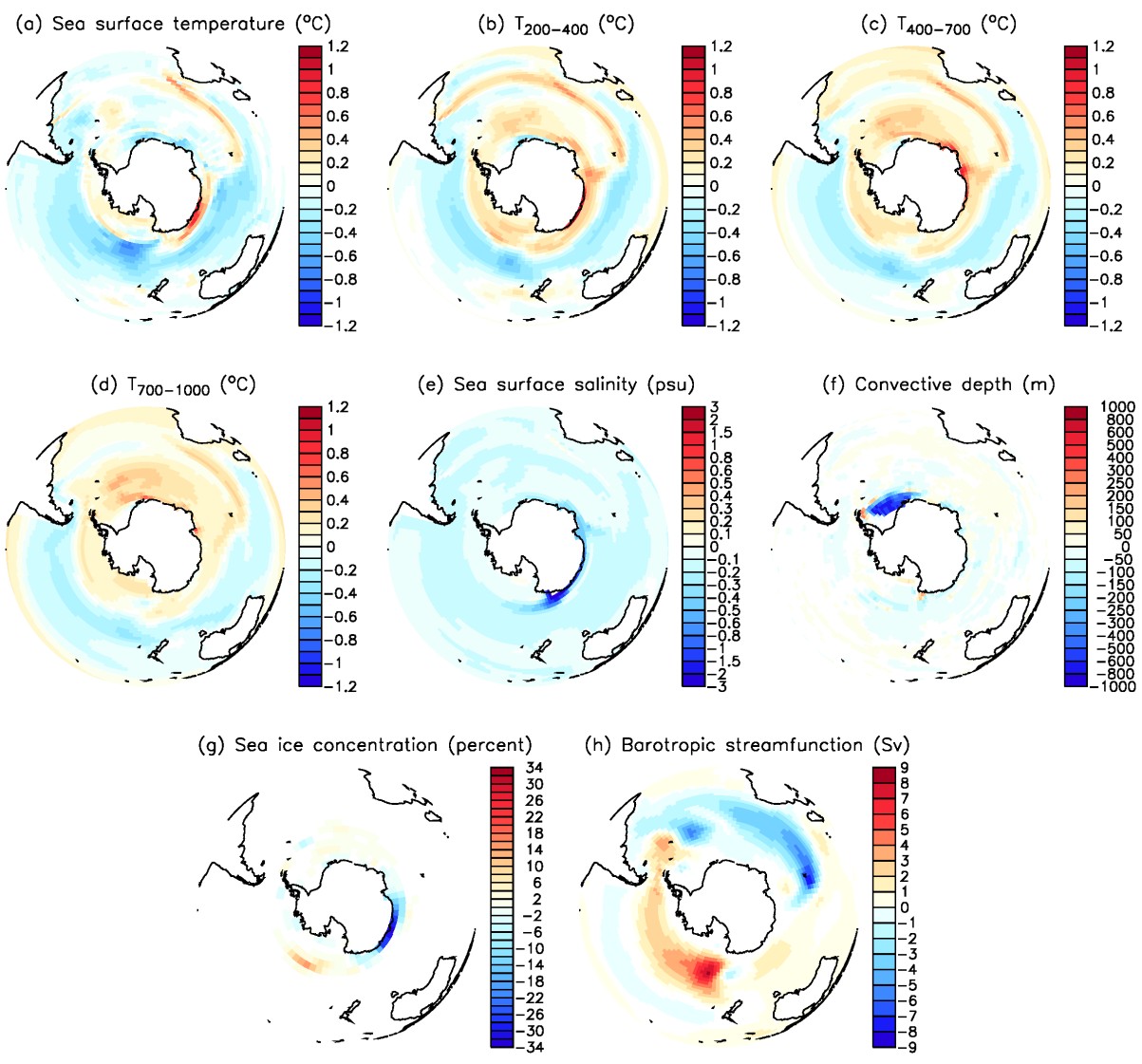

**Figure 5.** Mean anomalies for the final 100 years of the hosing phase (i.e. years 801–900) of experiment WILKES, relative to the equivalent years of the pre-industrial control simulation: **(a)** sea surface temperature (°C), **(b)** mean temperature at a depth of 200–400m (°C), **(c)** mean temperature at a depth of 400–700m (°C), **(d)** mean temperature at a depth of 700–1000m (°C), **(e)** sea surface salinity (psu), **(f)** convective depth (m), **(g)** sea ice concentration (%), and **(h)** barotropic streamfunction (Sv). Only values that are significant at the 5% probability level are shown.

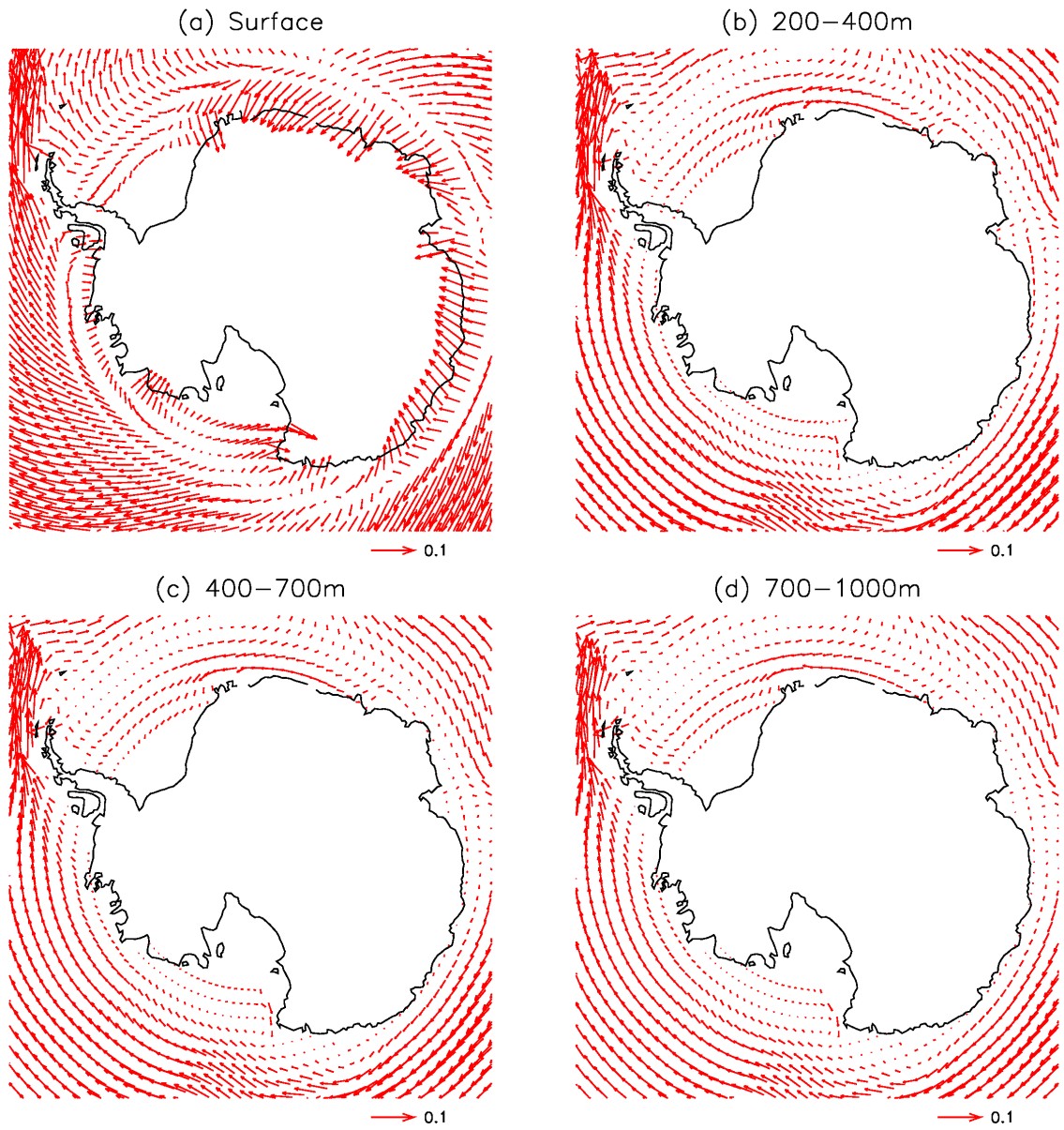

**Figure 6.** Climatological annual-mean ocean velocities $(\mathrm{ms}^{-1})$ for the pre-industrial control simulation: **(a)** surface velocity, **(b)** mean velocity at a depth of 200–400m, **(c)** mean velocity at a depth of 400–700m, and **(d)** mean velocity at a depth of 700–1000m.

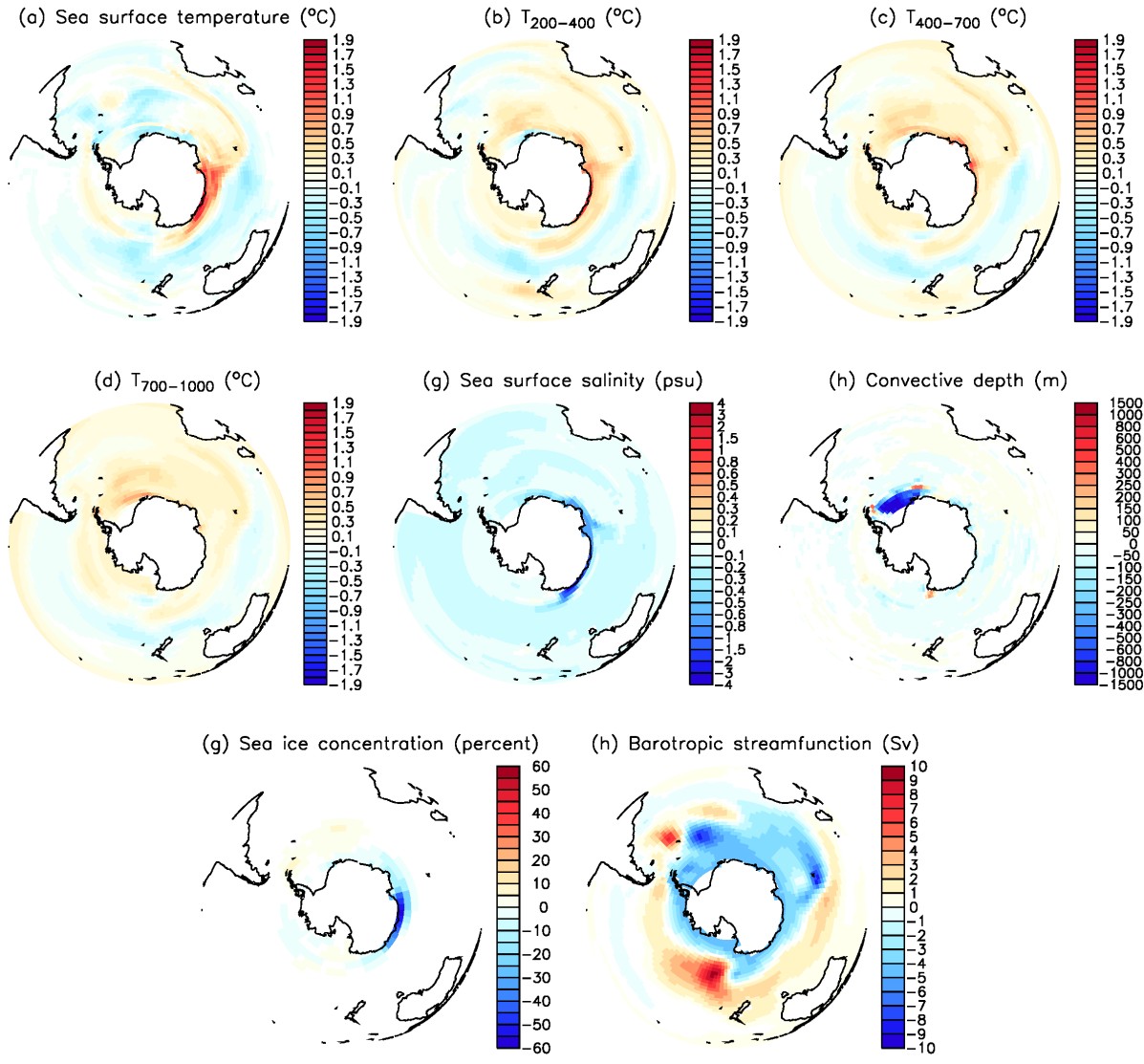

**Figure 7.** As Fig. 5, but for experiment WEST.

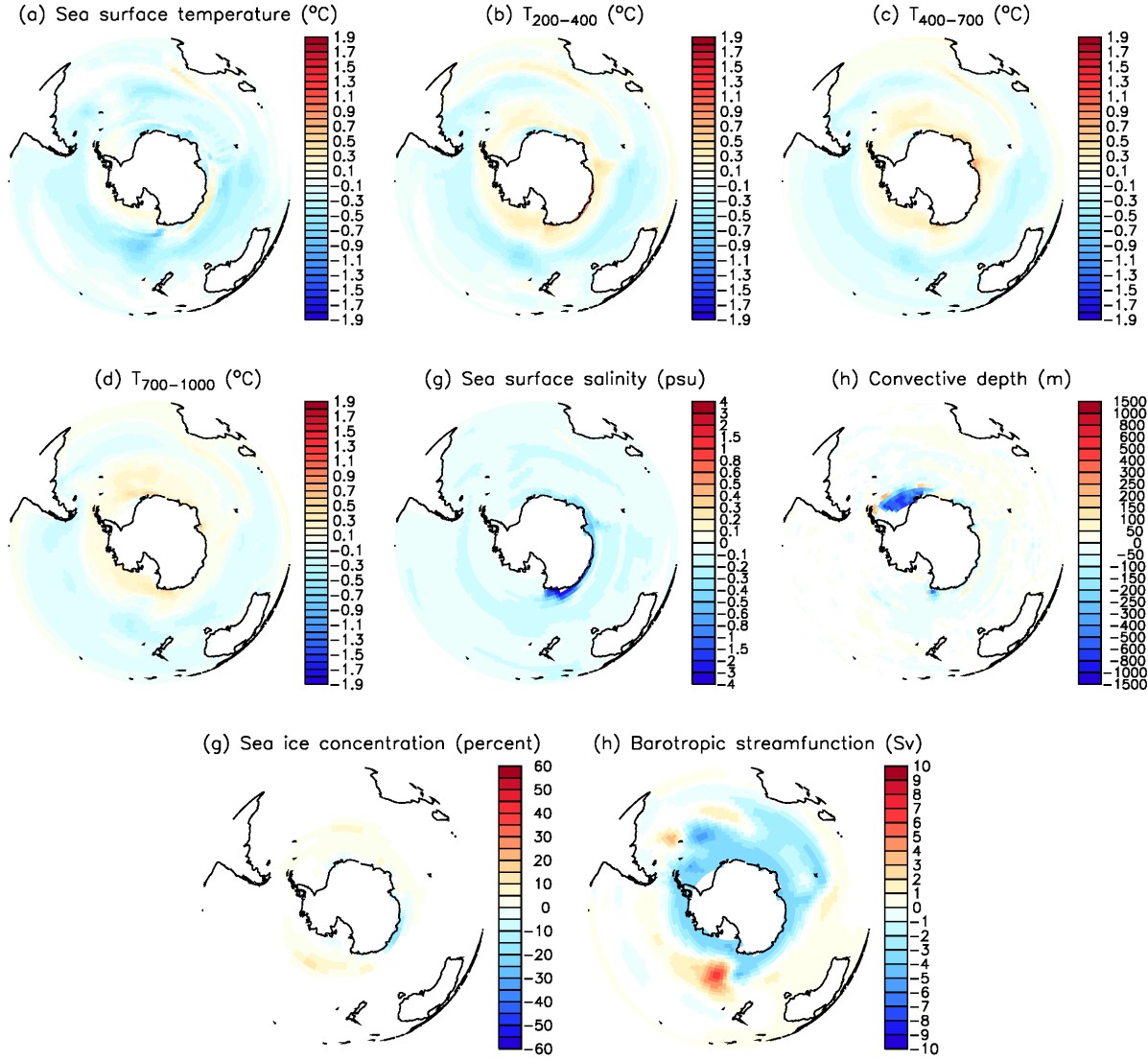

**Figure 8.** As Fig. 5, but for experiment EAST.

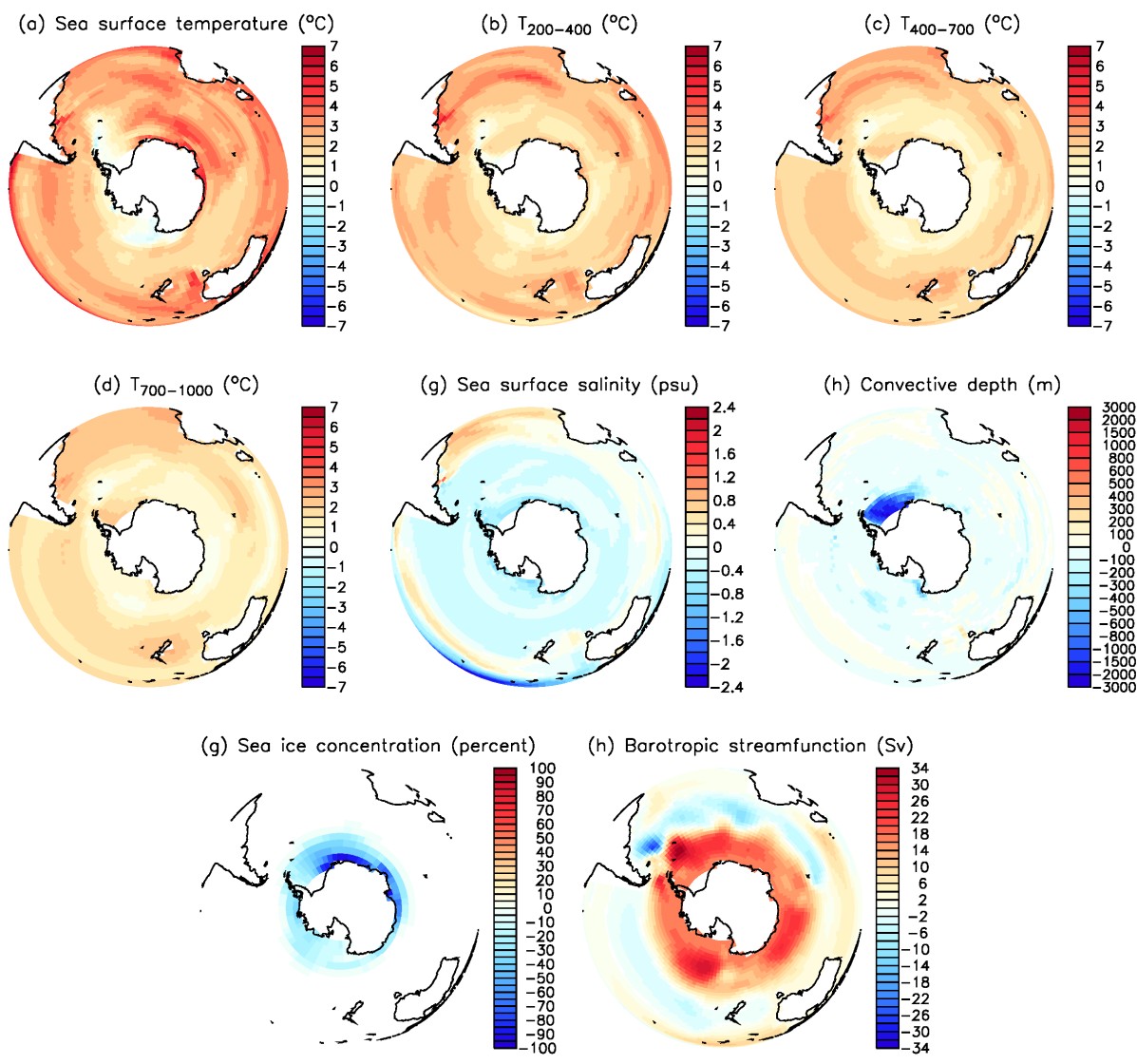

**Figure 9.** As Fig. 5, but for the first 100 years after the atmospheric $CO_2$ concentration has stabilised (i.e. years 141–240) of experiment 4CO2.