# Peer review of "Impacts of marine instability across the East Antarctic Ice Sheet on Southern Ocean dynamics"

_The Cryosphere, 2016_

## Short Comment (SC1) · 31 May 2016

Dear authors, this is a very interesting paper. Perhaps you are interested in this paper, even though it uses a very coarse ocean model, but it discusses the interaction of meltwater and Southern Ocean circulation Best wishes, Anders

T. Hattermann and A. Levermann Response of Southern Ocean circulation to global warming may enhance basal ice shelf melting around Antarctica Climate Dynamics 35 (2010), 741-756.
* * *

---

## Referee Comment (RC1) · Anonymous Referee #1 · 14 Jun 2016

The authors examine the effects of enhanced meltwater release from the Wilkes Basin (East Antarctic Ice Sheet, EAIS) on bottom water formation and dynamics in the Southern Ocean, based on an ensemble modeling approach using the CSIRO Mk3L climate system model. The ocean and the atmosphere model components both have a relative coarse horizontal resolution, ideal for performing long climate integrations but limited in capturing the processes relevant for deep and bottom water formation in the Southern Ocean marginal seas.

General comments: The paper represents a very timely study following recent claims that the EAIS might be as vulnerable to ocean warming as the West Antarctic Ice Sheet (WAIS) with significant contributions to global sea-level rise. This study now considers the inverse interaction, showing that enhanced meltwater release along the EAIS coast warms the seas further downstream, thus providing additional heat for melting at the

base of, e.g., Amery and Filchner-Ronne ice shelves; the latter also fed from the WAIS.

The contribution can be considered as a sensitivity study of interest for the climate modeling community but with limited relevance for the 'real' processes controlling today's deep and bottom water formation, which mainly occurs along the continental shelf break. I.e., the authors miss to emphasize that most of today's climate models have the tendency to form on decadal timescales a large Weddell polynya, in which most of the model's Antarctic Bottom Water (AABW) is formed. Therefore, the claimed reduction in AABW formation, due to a meltwater induced stabilization of the water column in the central Weddell Sea, might be just a model artifact.

Therefore, I urge the authors to consider their new findings carefully in view of the existing model limitations, but if done, I recommend publication in TC after consideration of the comments/ corrections listed below.

Specific comments: 1. The model setup bears several limitations with the model resolution already mentioned. However, the use of a constant atmospheric $CO_2$ concentration of 280 ppmV is another one. Especially in view of the stabilization of the water column in the Weddell Sea, a warmer and wetter atmosphere might be more efficient in hampering deep convection than meltwater advected thousands of kilometres and diluted by mixing and/or re-circulation as part of the gyre circulation.

2. The model results show a general warming of the water column downstream of the 'input sites' along the EAIS. While a warming of the deep layers can be explained by reduced deep convection, which brings warm waters to the surface and, in turn, cools the deep ocean, an explanation for the warming of the surface waters along the coast is missing. One can only speculate that this might be due to (a) a shorter lasting sea ice cover combined with an elongated period of solar heating of the surface waters and/or (2) the input of glacial melt at 0 degC instead of roughly -2 degC. Please provide a plausible explanation.

Technical corrections P5/L02: First time, but throughout the paper: The term 'coastal

counter-currents' is unusual. Please use 'Antarctic slope current' or 'coastal current' instead. P5/L10: The greatest reduction in convective depth occurs in the western Weddell Sea. Note that the Weddell Sea extends at least up to 30o E. P5/L19: "Consistent with the greater decrease in AABW formation, . . .."

---

## Referee Comment (RC2) · Anonymous Referee #2 · 22 Jun 2016

This paper explores the impact of freshwater input from the Wilkes Basin on the broader Antarctic ice sheet and Southern Ocean. In particular the potential for loss of ice in the Wilkes Basin to trigger warming and further ice loss in the Weddell Sea sector is discussed. It builds on the work of Fogwill et al., (2015), with this paper differing in that freshwater hosing is applied to the East Antarctic as opposed to the West Antarctic. The paper is important given recent publications highlighting the sensitivity of the East Antarctic ice sheet to future anthropogenic climate change, but which lack potential ocean feedback effects. The paper is well written and the discussion and conclusions are justified based on the results presented. I recommend publication once minor comments below are addressed.

Specific Comments:

[Figure]

1. The WEST/EAST/WILKES experiments are useful in highlighting the sensitivity of the model to the area that freshwater hosing occurs. In the discussion you could also touch on the assumption made that all ice lost from the Wilkes Basin would lead to localized meltwater delivery to the immediate George V Coast. As discussed there are various proposed mechanisms for collapse of the Wilkes Basin, which may affect the validity of this assumption.

2. I agree that using fixed pre-industrial $CO_2$ makes sense for the purposes of a sensitivity test in order to isolate other effects. However around lines 30 on page 3 could you expand on the impact that elevated $CO_2$ may have on these results, given that elevated $CO_2$ is likely required to trigger a collapse of the Wilkes Basin. Ideally additional experiments would have been performed at elevated $CO_2$ with and without freshwater hosing.

3. The a) and b) panels of Figures 5, 7 and 8 suggest that there would be surface cooling in the Weddell Sea sector and potentially at depths of 200-400m. This switches to warming at depth, as discussed in the paper. Although a possible mechanism that could explain this surface cooling is discussed, the surface cooling in the Weddell Sea sector is not currently mentioned in the manuscript. Could you also include some discussion about how warming/cooling at different ocean depths may affect ice sheet stability in the Weddell Sea sector.

Technical Corrections:

Page 4, line 5, could also include reference to Bintanja et al., 2013.

---

## Author Comment (AC1) · 21 Jul 2016

**Response to Short Comment 1: 'Southern Subpolar Gyre changes' by Anders Levermann**

*Dear authors, this is a very interesting paper. Perhaps you are interested in this paper, even though it uses a very coarse ocean model, but it discusses the interaction of meltwater and Southern Ocean circulation Best wishes, Anders*

*T. Hattermann and A. Levermann Response of Southern Ocean circulation to global warming may enhance basal ice shelf melting around Antarctica Climate Dynamics 35 (2010), 741-756.*

We thank Anders Levermann for his kind comments.

Hattermann and Levermann (2010) use an intermediate complexity model to study the response of the Southern Ocean (SO) to increasing atmospheric $CO_2$. A simple parameterisation is used to represent the heat and freshwater fluxes due to basal melting of the Antarctic ice shelves. The authors find that a strengthening of the Antarctic Circumpolar Current (ACC) leads to warming of the SO and hence enhanced basal melting. They also identify three feedback loops whereby basal melting influences the circulation of the SO. The dominant feedback loop involves local cooling; this loop is negative. A secondary feedback loop involves acceleration of the subpolar gyres; this loop is positive. A minor additional feedback loop involves weakening of the ACC; this loop is negative.

This study is extremely pertinent, and we thank Anders for bringing it to our attention. We will incorporate discussion of it into the manuscript and, where appropriate, we will also test for the presence of each of the three feedback loops within our experiments.

We have also completed an additional experiment in which we increase the atmospheric $CO_2$ concentration to four times the pre-industrial concentration (1120 ppm) at 1% per year. We will incorporate this experiment into the manuscript. This will allow us to compare and contrast the response to increased freshwater input with the response to increasing atmospheric $CO_2$. It will also allow us to compare our experiments with those conducted by Hattermann and Levermann (2010).

---

## Author Comment (AC2) · 21 Jul 2016

**Response to Referee Comment 1: 'Reviewer comment' by Anonymous Referee #1**

*The authors examine the effects of enhanced meltwater release from the Wilkes Basin (East Antarctic Ice Sheet, EAIS) on bottom water formation and dynamics in the Southern Ocean, based on an ensemble modeling approach using the CSIRO Mk3L climate system model. The ocean and the atmosphere model components both have a relative coarse horizontal resolution, ideal for performing long climate integrations but limited in capturing the processes relevant for deep and bottom water formation in the Southern Ocean marginal seas.*

We thank the referee for positive and constructive feedback on the manuscript.

[Figure]

*General comments: The paper represents a very timely study following recent claims that the EAIS might be as vulnerable to ocean warming as theWest Antarctic Ice Sheet (WAIS) with significant contributions to global sea-level rise. This study now considers the inverse interaction, showing that enhanced meltwater release along the EAIS coast warms the seas further downstream, thus providing additional heat for melting at the base of, e.g., Amery and Filchner-Ronne ice shelves; the latter also fed from the WAIS.*

*The contribution can be considered as a sensitivity study of interest for the climate modeling community but with limited relevance for the 'real' processes controlling to-day's deep and bottom water formation, which mainly occurs along the continental shelf break. I.e., the authors miss to emphasize that most of today's climate models have the tendency to form on decadal timescales a large Weddell polynya, in which most of the model's Antarctic Bottom Water (AABW) is formed. Therefore, the claimed reduction in AABW formation, due to a meltwater induced stabilization of the water column in the central Weddell Sea, might be just a model artifact.*

*Therefore, I urge the authors to consider their new findings carefully in view of the existing model limitations, but if done, I recommend publication in TC after consideration of the comments/ corrections listed below.*

We acknowledge that the relatively coarse spatial resolution of the model has both advantages and disadvantages, as stated clearly by the referee. In particular, we acknowledge that the formation of Antarctic Bottom Water within the model occurs in the open Weddell Sea and not along the continental shelf.

We will revise the manuscript to include a discussion of these issues. This discussion will acknowledge that our conclusions may be model-dependent, and will emphasise the importance of conducting similar experiments using models with a higher spatial resolution.

*Specific comments: 1. The model setup bears several limitations with the model resolution already mentioned. However, the use of a constant atmospheric $CO_2$ concen-*

*tration of 280 ppmV is another one. Especially in view of the stabilization of the water column in the Weddell Sea, a warmer and wetter atmosphere might be more efficient in hampering deep convection than meltwater advected thousands of kilometres and diluted by mixing and/or re-circulation as part of the gyre circulation.*

We have completed an additional experiment in which we increase the atmospheric $CO_2$ concentration to four times the pre-industrial concentration (1120 ppm) at 1% per year. We will incorporate this experiment into the manuscript. This will allow us to compare and contrast the response to increased freshwater input with the response to increasing atmospheric $CO_2$.

*2. The model results show a general warming of the water column downstream of the 'input sites' along the EAIS. While a warming of the deep layers can be explained by reduced deep convection, which brings warm waters to the surface and, in turn, cools the deep ocean, an explanation for the warming of the surface waters along the coast is missing. One can only speculate that this might be due to (a) a shorter lasting sea ice cover combined with an elongated period of solar heating of the surface waters and/or (2) the input of glacial melt at 0 degC instead of roughly -2 degC. Please provide a plausible explanation.*

We have examined the sea ice changes within our experiments and we can confirm that there is reduced sea ice cover within the regions where there is warming of the surface waters. We will incorporate this analysis into the manuscript.

There is no heat flux associated with the freshwater input: the water that is added to the ocean is effectively assumed to have the same temperature as the ambient seawater. We will revise the text of the manuscript to clarify this point.

*Technical corrections P5/L02: First time, but throughout the paper: The term 'coastal counter-currents' is unusual. Please use 'Antarctic slope current' or 'coastal current' instead. P5/L10: The greatest reduction in convective depth occurs in the western Weddell Sea. Note that the Weddell Sea extends at least up to 30o E. P5/L19: "Con-*

*sistent with the greater decrease in AABW formation, . . ..”*

P5/L02: We will revise the text to use the term 'coastal current' instead.

P5/L10: We will clarify all statements that refer to locations within the Weddell Sea.

P5/L19: We will correct this typographical error.

---

## Author Comment (AC3) · 21 Jul 2016

**Response to Referee Comment 2: 'Reviewer Comment' by Anonymous Referee #2**

*This paper explores the impact of freshwater input from the Wilkes Basin on the broader Antarctic ice sheet and Southern Ocean. In particular the potential for loss of ice in the Wilkes Basin to trigger warming and further ice loss in the Weddell Sea sector is discussed. It builds on the work of Fogwill et al., (2015), with this paper differing in that freshwater hosing is applied to the East Antarctic as opposed to the West Antarctic. The paper is important given recent publications highlighting the sensitivity of the East Antarctic ice sheet to future anthropogenic climate change, but which lack potential ocean feedback effects. The paper is well written and the discussion and conclusions*

*are justified based on the results presented. I recommend publication once minor comments below are addressed.*

We thank the referee for positive and constructive feedback on the manuscript.

*Specific Comments:*

*1. The WEST/EAST/WILKES experiments are useful in highlighting the sensitivity of the model to the area that freshwater hosing occurs. In the discussion you could also touch on the assumption made that all ice lost from the Wilkes Basin would lead to localized meltwater delivery to the immediate George V Coast. As discussed there are various proposed mechanisms for collapse of the Wilkes Basin, which may affect the validity of this assumption.*

We will revise the manuscript so that this assumption is clearly stated. We will also include a discussion of proposed mechanisms for the collapse of the Wilkes Basin, and include an explicit acknowledgement that our assumption may not be entirely valid.

*2. I agree that using fixed pre-industrial CO2 makes sense for the purposes of a sensitivity test in order to isolate other effects. However around lines 30 on page 3 could you expand on the impact that elevated CO2 may have on these results, given that elevated CO2 is likely required to trigger a collapse of the Wilkes Basin. Ideally additional experiments would have been performed at elevated CO2 with and without freshwater hosing.*

We have completed an additional experiment in which we increase the atmospheric $CO_2$ concentration to four times the pre-industrial concentration (1120 ppm) at 1% per year. We will incorporate this experiment into the manuscript. This will allow us to compare and contrast the response to increased freshwater input with the response to increasing atmospheric $CO_2$.

*3. The a) and b) panels of Figures 5, 7 and 8 suggest that there would be surface cooling in the Weddell Sea sector and potentially at depths of 200-400m. This switches*

*to warming at depth, as discussed in the paper. Although a possible mechanism that could explain this surface cooling is discussed, the surface cooling in the Weddell Sea sector is not currently mentioned in the manuscript. Could you also include some discussion about how warming/cooling at different ocean depths may affect ice sheet stability in the Weddell Sea sector.*

The surface cooling, in conjunction with warming at depth, is a consequence of reduced vertical mixing. Using the same climate model as us, this mechanism is studied by Fogwill et al. (2015). We will incorporate a discussion of the temperature changes in the Weddell Sea sector into the manuscript. This discussion will cover the underlying mechanisms and will address the consequences for ice sheet stability.

Fogwill, C. J., S. J. Phipps, C. S. M. Turney and N. R. Golledge: Sensitivity of the Southern Ocean to enhanced regional Antarctic ice sheet meltwater input, *Earth's Future*, 3, 317–329, doi:10.1002/2015EF000306, 2015.

*Technical Corrections:*

*Page 4, line 5, could also include reference to Bintanja et al., 2013.*

We will include a reference to this study:

Bintanja, R., G. J. van Oldenborgh, S. S. Drijfhout, B. Wouters and C. A. Katsman: Important role for ocean warming and increased ice-shelf melt in Antarctic sea-ice expansion, *Nature Geoscience*, 6, 376–379, doi:10.1038/NGEO1767, 2013.

We have also examined the sea ice changes within our experiments, and we will incorporate this analysis into the manuscript.